# Clinical progression and outcomes of patients hospitalized with COVID-19 in humanitarian settings: A prospective cohort study in South Sudan and Eastern Democratic Republic of the Congo

**Shannon Doocy**[1]*, **Iris Bollemeijer**[2], **Eva Leidman**[1,3], **Abdou Sebushishe**[4], **Eta Ngole Mbong**[5], **Kathleen Page**[6]

1 Department of International Health, Johns Hopkins University Bloomberg School of Public Health, Baltimore, Maryland, United States of America, 2 International Medical Corps–Santa Monica, California, United States of America, 3 US Centers for Disease Control and Prevention, Atlanta, Georgia, United States of America, 4 International Medical Corps–Juba, South Sudan, 5 International Medical Corps–Goma, North Kivu, Democratic Republic of the Congo, 6 Johns Hopkins University School of Medicine, Baltimore, Maryland, United States of America

* doocy1@jhu.edu

**Data Availability Statement:** Our data set is available on Humanitarian Data Exchange: https://

## Abstract

Little information is available on COVID-19 in Africa and virtually none is from humanitarian and more resource-constrained settings. This study characterizes hospitalized patients in the African humanitarian contexts of Juba, South Sudan and North and South Kivu in Eastern Democratic Republic of the Congo. This observational cohort was conducted between December 2020 and June 2021. Patients presenting for care at five facilities or referred from home-based care by mobile medical teams were eligible for enrollment and followed until death or recovery. Disease progression was characterized for hospitalized patients using survival analysis and mixed effects regression model to estimate survival odds for patient characteristics and treatments received. 144 COVID-19 cases enrolled as hospitalized patients were followed to recovery/death. The observed mortality proportion among hospitalized patients was 16.7% (CI: 11.2–23.3%); mortality was three times higher in South Sudan, where patients presented later after symptom onset and in worse conditions. Age and diabetes history were the only patient characteristics associated with decreased survival; clinical status indicators associated with decreased survival included fever, low oxygen level, elevated respiratory and pulse rates. The only therapy associated with survival was non-invasive oxygen; invasive oxygen therapies and other specialized treatments were rarely received. Improving availability of oxygen monitoring and proven COVID-19 therapies in humanitarian and resource-poor settings is critical for health equity. Customizing training to reflect availability of specific medications, therapies and operational constraints is particularly important given the range of challenges faced by providers in these settings.

data.humdata.org/dataset/clinical-progression-and-outcomes-of-patients-hospitalized-with-covid-19-in-humanitarian-settings.

**Funding:** This study was funded by the United States Agency for International Development in the form of a grant (72OFDA20GR0221) awarded to SD. The funders had no role in study design, data collection and analysis, decision to publish, or preparation of the manuscript.

**Competing interests:** The authors declare no competing interests.

## Introduction

COVID-19 has caused 4.7 million deaths globally and available data suggests a lesser impact in Africa, where 5.9 million cases and ~143,000 deaths were reported as of September, 2021 [1] Africa accounts for 14% of the global population and <3% of reported COVID-19 cases and deaths [1, 2]. While younger population structure and limited testing capacity likely contribute to lower case counts and mortality, the impact of COVID-19 is almost certainly underreported, in particular considering evidence from seroprevalence surveys which suggest that 38% of the population in Juba, South Sudan and 41% of the population in Bukavu, a city in eastern Democratic Republic of the Congo, had been infected with COVID-19 by late 2020 [3, 4] Given slow vaccine rollout and limited health infrastructure, the continent's 1.2 billion residents continue to face tremendous risk of infection as the COVID-19 pandemic continues.

Most African countries have low vaccination coverage, inadequate diagnostic and laboratory capacity, and at hospitals limited staffing and availability of evidence-based COVID-19 treatments such as ventilators, antivirals and monoclonal antibodies hamper quality of care. Furthermore, tertiary facilities are often overcrowded and difficult to access, particularly for rural populations. Sub-Saharan Africa has 1.2 hospital beds per 1000 population compared to 2.3/1000 in all low and middle-income countries; similarly, sub-Saharan Africa has 0.2 physicians/1000 compared to 1.3/1000 in low and middle-income countries [5, 6]. While there is great variation in capacity across the continent, service availability is particularly limited in the lowest income countries, many of which have a history of protracted conflict. In South Sudan (SSD), there are 0.15 physicians/1000, and a recent health system assessment indicated poor quality of care (estimates of hospital bed per capita were not available) [7, 8]. Similarly, Democratic Republic of the Congo (DRC) has only 0.1 physicians/1000 and 0.8 hospital beds/1000, which are among the lowest in the world. In both countries, health system capacity and access are inadequate [5].

Relatively little is known about the profile and outcomes of hospitalized COVID-19 patients in Africa compared to other settings. The recent African COVID-19 Critical Care Outcomes Study (ACCCOS) study enrolled >3,750 inpatients in ten African countries and estimated the hospital mortality proportion at 48.2%, much higher than rates observed in European, Asian, and American hospitals; elevated mortality was attributed to insufficient critical care resources [9]. This paper characterizes clinical progression and outcomes of hospitalized COVID-19 patients in Juba, South Sudan and North and South Kivu, Eastern DRC and aims to inform the COVID-19 response in resource-poor and conflict-affected settings in Africa.

## Methods

A prospective observational cohort of COVID-19 cases was conducted between December 2020 and June 2021 in five health facilities operated or supported by International Medical Corps (IMC) in Eastern DRC (n = 4) and Juba, South Sudan (n = 1), including four that provided inpatient care. Hospital characteristics and COVID-19 related care capacity among the four facilities providing inpatient care are summarized in **S1 Table**; laboratory and diagnostic testing, oxygen therapies and available medications varied across facilities. Both COVID-19 cases receiving inpatient and outpatient care were enrolled in the study with the primary aim of identifying risk factors for poor outcomes, including hospitalization and death as described in the companion paper [10] a secondary aim was to document clinical progression and characterize COVID-19 clinical management. Sample size calculations were conducted based on the primary aim and are presented in the risk factor paper; due to the descriptive nature of this paper, where detecting differences between countries or hypothesis testing was not an aim, additional sample size calculations were not conducted. Individuals presenting for care at a

study facility or referred by mobile medical teams providing home-based care in the facility catchment area with a positive RT-PCR or antigen test and inpatients not tested meeting the national suspect case definitions were eligible for enrollment. In DRC, a case met the syndromic case definition if they had one or more of the following sign(s) or symptom(s): fever, dry cough, headache, severe fatigue, sore throat, shortness of breath, dyspnea (difficulty breathing), muscle or joint pain, or coryza (common cold). In South Sudan, suspect cases presented with acute onset of fever ≥38˚C and cough, or an acute onset of any three or more signs or symptoms, including those in the DRC case definition as well as anorexia, nausea, vomiting, diarrhea, and altered mental status. Cases were subsequently excluded from analysis if they tested negative following enrollment or were transferred to another facility for care. Cases treated as inpatient were considered recovered if they were discharged alive from inpatient care. All eligible cases (n = 751) were recruited of which 592 consented to participate and were enrolled and 519 were followed to recovery or death, including 144 patients hospitalized at four health facilities which are the focus of this paper (Fig 1).

Oral consent was obtained from adults and parental consent for subjects <18 years by trained research nurses. A questionnaire-based interview was conducted by IMC research nurses or Ministry of Health facility staff at each study facility trained by the lead investigator in each country. The interview, including demographic, symptoms and health history was conducted along with direct observation of mid-upper arm circumference (MUAC), weight, height, pulse rate, oxygen saturation and hemoglobin levels using Masimo RAD 57 (Masimo, Irvine CA), the Multi-Parameter Patient Monitor YK8000 (Yonker, Jiangsu, China) and

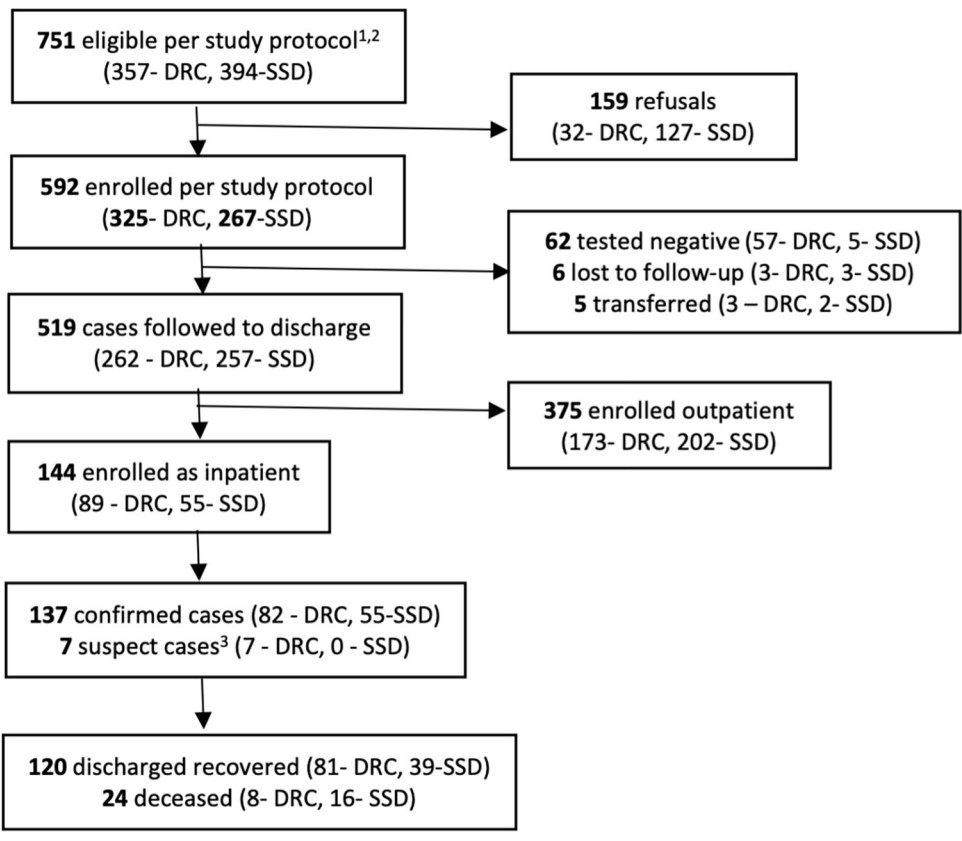

**Fig 1. Study participant flow diagram.**

HemoCue 301 devices (HemoCue, Ängelholm, Sweden); devices varied due to availability but are comparable [11, 12]. Malaria rapid diagnostic tests (Malaria Ag, SD) were available if providers suspected malaria and Hemoglobin A1c (HbA1c) was measured with A1C Now+ Professional Test Kits (PTS Diagnostics, Whitestown, IN) for patients with self-reported diabetes history. Inpatients were intended to have daily follow-ups with clinical course data collected by either health facility staff or research nurses. Participants were followed until discharge (considered as recovery), death, or transfer to a different facility.

In SSD, data entry was direct into the CommCare Platform (Dimagi, Cambridge MA); in DRC data was recorded on paper and subsequently entered electronically. Data was verified by research managers prior to further real-time review for quality and completeness. Data analysis was conducted in R version 4.0.4 (RStudio, PBC, Boston MA). Nutrition status was classified using WHO body mass index (BMI) cut-offs for ages $\geq$19 years and BMI-for-age for 5–19 years [13, 14] anemia was defined using WHO age/sex specific thresholds for hemoglobin concentration [15]. Definitions and cutoffs for other clinical parameters, were developed based on clinician consensus and clinical practice guidelines. Descriptive analysis included a comparison of continuous variables using Kruskal-Wallis test and categorical variables using Fisher's exact test. Mixed effects logistic regression models were used to evaluate odds of mortality using parameters significant at $p<0.1$ in unadjusted models; models were adjusted for age, sex, country, and nationality as fixed effects and facility as a random effect. Survival analysis evaluated time from self-reported symptom onset by country and oxygen levels at enrollment using Kaplan-Meier survival functions; survival functions are unadjusted for severity and were evaluated with a log rank test.

The study was reviewed and approved by the Johns Hopkins University Institutional Review Board, the South Sudan Ministry of Health Ethics Committee, the University of Kinshasa School of Public Health, and the United States Centers for Disease Control and Prevention (US CDC). The study is registered with ClinicalTrials.gov (NCT04568499) and was funded by USAID (award 72OFDA20GR0221).

## Results

The study included 519 COVID-19 cases followed to discharge, including 144 (27.7%) hospitalized patients who are the focus of this analysis. The hospital mortality proportion was 16.7% (CI: 11.2–23.3%) and was significantly greater in SSD than DRC (29.1% vs. 9.0%; p = 0.003). Patient characteristics differed by country, where SSD had a significantly larger proportion than DRC of inpatients who were non-national (26.4% vs. 3.4%, p<0.001) and male (72.7% vs. 56.2%, p = 0.046); there was no significant difference in age or medical history. The most frequent self-reported symptoms at admission were cough (71.5%), fatigue (61.1%), headache (53.8%), shortness of breath (52.1%) and chest pain (43.4%). Of note, fever was a less common symptom than anticipated (observed in 19% and self-reported by 46% of patients at enrollment). Patients in SSD presented in worse condition by various clinical parameters, likely a function of greater duration from symptom onset to study enrollment (10.2 vs. 6.1 days, p<0.001) (**Table 1**).

When examined by survival status, there was no significant difference by sex, however, deceased patients were significantly older (61.2 vs. 45.3 years, p<0.001) and more likely to report symptoms. The only significant difference in medical history between deceased and surviving patients was prior diabetes diagnosis (45.8% vs. 14.2%, p<0.001) (Table 2). Differences in survival probability by day since symptom onset are presented in **Fig 2**. While the mortality proportion differed significantly between SSD and DRC overall, there was no difference in survival probability when adjusted for the date of symptom onset. Oxygen saturation at admission

**Table 1. Demographic characteristics and self-reported symptoms and health history at enrollment.**

| | All Inpatients | By Enrollment Country | | | By Outcome | | |
|---|---|---|---|---|---|---|---|
| | | DR Congo | S Sudan | p-value | Deceased | Recovered | p-value |
| | N = 144 | N = 89 | N = 55 | | N = 24 | N = 120 | |
| **Days from Symptom Onset to Enrollment** (mean, SD) | 7.8 (7.4) | 6.1 (5.7) | 10.2 (8.8) | <0.001 | 8.3 (5.4) | 7.7 (7.8) | 0.27 |
| **Time in Study** (mean, SD) | 9.2 (6.0) | 7.9 (4.3) | 11.3 (7.6) | **0.004** | 4. 5 (4.6) | 10.2 (5.8) | <0.001 |
| **Age in years** (mean, SD) | 48.0 (18.8) | 45.7 (20.5) | 51.6 (15.2) | 0.086 | 61.2 (11.7) | 45.3 (18.9) | <0.001 |
| **Age categories** < 18 | 5 (3.5%) | 5 (5.6%) | 0 (0.0%) | 0.29 | 0 (0.0%) | 5 (4.2%) | **0.003** |
| 18–44 | 57 (39.6%) | 37 (41.6%) | 20 (36.4%) | | 3 (12.5%) | 54 (45.0%) | |
| 45–64 | 50 (34.7%) | 29 (32.6%) | 21 (38.2%) | | 10 (41.7%) | 40 (33.3%) | |
| 65+ | 32 (22.2%) | 18 (20.2%) | 14 (25.5%) | | 11 (45.8%) | 21 (17.5%) | |
| **Sex** Female | 54 (37.5%) | 39 (43.8%) | 15 (27.3%) | **0.046** | 6 (25.0%) | 48 (40.0%) | 0.170 |
| Male | 90 (62.5%) | 50 (56.2%) | 40 (72.7%) | | 18 (75.0%) | 72 (60.0%) | |
| **Nationality** National | 125 (88.0%) | 86 (96.6%) | 39 (73.6%) | <0.001 | 22 (91.7%) | 103 (87.3%) | 0.74 |
| Non-National | 17 (12.0%) | 3 (3.4%) | 14 (26.4%) | | 2 (8.3%) | 15 (12.7%) | |
| **Symptoms (self-report)** | | | | | | | |
| Any symptom(s) | 129 (89.6%) | 76 (85.4%) | 53 (96.4%) | **0.036** | 24 (100.0%) | 105 (87.5%) | 0.076 |
| Cough | 103 (71.5%) | 54 (60.7%) | 49 (89.1%) | <0.001 | 20 (83.3%) | 83 (69.2%) | 0.16 |
| Fatigue/ malaise | 88 (61.1%) | 50 (56.2%) | 38 (69.1%) | 0.12 | 20 (83.3%) | 68 (56.7%) | **0.014** |
| Headache | 77 (53.8%) | 41 (46.6%) | 36 (65.5%) | **0.028** | 11 (45.8%) | 66 (55.5%) | 0.39 |
| Shortness of breath | 75 (52.1%) | 33 (37.1%) | 42 (76.4%) | <0.001 | 22 (91.7%) | 53 (44.2%) | <0.001 |
| Chest pain | 62 (43.4%) | 27 (30.7%) | 35 (63.6%) | <0.001 | 17 (70.8%) | 45 (37.8%) | **0.003** |
| Runny nose | 54 (37.8%) | 27 (30.7%) | 27 (49.1%) | **0.027** | 5 (21.7%) | 49 (40.8%) | 0.084 |
| Sore throat | 52 (36.4%) | 16 (18.2%) | 36 (65.5%) | <0.001 | 8 (33.3%) | 44 (37.0%) | 0.74 |
| Muscle/ joint pain | 50 (35.0%) | 24 (27.3%) | 26 (47.3%) | **0.015** | 9 (37.5%) | 41 (34.5%) | 0.78 |
| Loss of taste/ smell | 25 (17.5%) | 11 (12.5%) | 14 (25.5%) | **0.047** | 8 (33.3%) | 17 (14.3%) | **0.037** |
| Chills | 24 (16.7%) | 16 (18.0%) | 8 (14.5%) | 0.59 | 3 (12.5%) | 21 (17.5%) | 0.77 |
| Abdominal pain | 24 (16.7%) | 12 (13.5%) | 12 (21.8%) | 0.19 | 1 (4.2%) | 23 (19.2%) | 0.079 |
| Vomiting/nausea | 22 (15.3%) | 16 (18.0%) | 6 (10.9%) | 0.25 | 3 (12.5%) | 19 (15.8%) | >0.99 |
| Wheezing | 13 (9.0%) | 8 (9.0%) | 5 (9.1%) | >0.99 | 5 (20.8%) | 8 (6.7%) | **0.043** |
| Diarrhea | 10 (6.9%) | 4 (4.5%) | 6 (10.9%) | 0.18 | 1 (4.2%) | 9 (7.5%) | >0.99 |
| Loss of appetite | 8 (5.6%) | 3 (3.4%) | 5 (9.1%) | 0.26 | 4 (16.7%) | 4 (3.3%) | **0.027** |
| **Medical History (self-report)** | | | | | | | |
| BCG vaccine | 120 (83.3%) | 78 (87.6%) | 42 (76.4%) | 0.078 | 18 (75.0%) | 102 (85.0%) | 0.24 |
| Tuberculosis (prior) | 2 (1.4%) | 1 (1.1%) | 1 (1.8%) | >0.99 | 1 (4.2%) | 1 (0.8%) | 0.31 |
| HIV positive (n = 66) | 2 (3.0%) | 2 (6.5%) | 0 (0.0%) | 0.22 | 1 (7.7%) | 1 (1.9%) | 0.36 |
| Diabetes | 28 (19.4%) | 16 (18.0%) | 12 (21.8%) | 0.57 | 11 (45.8%) | 17 (14.2%) | **0.001** |
| Chronic Cardiac Disease | 11 (7.9%) | 9 (10.3%) | 2 (3.8%) | 0.21 | 0 (0.0%) | 11 (9.4%) | 0.21 |
| Chronic Pumonary Disease | 3 (2.1%) | 3 (3.4%) | 0 (0.0%) | 0.29 | 1 (4.3%) | 2 (1.7%) | 0.42 |
| Asthma | 2 (1.4%) | 1 (1.1%) | 1 (1.9%) | >0.99 | 0 (0.0%) | 2 (1.7%) | >0.99 |
| Current smoker | 3 (2.1%) | 3 (3.4%) | 0 (0.0%) | 0.30 | 0 (0.0%) | 3 (2.5%) | >0.99 |

was related to survival, where oxygen saturation ≥94% at admission had a five-day survival probability of 0.96 compared to 0.71 among patients with oxygen saturation <94% (p<0.001).

Patients in SSD were significantly more likely to receive oxygen support (56.4% vs. 29.2%, p = 0.001), less likely to be placed in a prone or half-seated position (60.0% vs. 100.0%, p<0.001) and more likely to receive medications within 24 hours of admission (**Table 2**). Laboratory and diagnostic tests were not assessed due to the small sample size, and are described for the entire study population. Blood samples were collected for 38.2% (n = 55) patients;

**Table 2. Clinical observations, therapies and medications at admission (within 24 hours of admission).**

| Clinical Observations | All Inpatients | By Enrollment Country | | | By Outcome | | |
|---|---|---|---|---|---|---|---|
| | | DR Congo | S Sudan | p-value | Deceased | Recovered | p-value |
| | N = 144 | N = 89 | N = 55 | | N = 24 | N = 120 | |
| Healthy appearance | 42 (29.2%) | 25 (28.1%) | 17 (30.9%) | 0.90 | 0 (0.0%) | 42 (35.0%) | <0.001 |
| Acutely ill (ambulatory) | 23 (16.0%) | 15 (16.9%) | 8 (14.5%) | | 0 (0.0%) | 23 (19.2%) | |
| Acutely ill (not ambulatory) | 79 (54.9%) | 49 (55.1%) | 30 (54.5%) | | 24 (100.0%) | 55 (45.8%) | |
| Mean temp (C) (SD) | 36.8 (0.9) | 36.7 (0.8) | 37.0 (0.9) | 0.006 | 37.1 (1.2) | 36.8 (0.8) | 0.27 |
| Fever (>37.5 C) | 27 (18.8) | 14 (15.7) | 13 (23.6) | 0.24 | 8 (33.3) | 19 (15.8) | 0.08 |
| Mean arterial pressure (SD) | 96.9 (13.9) | 95.5 (14.2) | 99.0 (13.1) | 0.12 | 101.3 (15.9) | 96.0 (13.3) | 0.15 |
| Hypotension (<90 SBP / <60 DBP)[1] | 6 (4.3%) | 5 (5.8%) | 1 (1.8%) | 0.40 | 1 (4.2%) | 5 (4.3%) | >0.99 |
| Hypertension (>140 SBP / >90 DBP)[1] | 20 (14.2%) | 12 (14.0%) | 8 (14.5%) | 0.92 | 5 (20.8%) | 15 (12.8%) | 0.34 |
| Mean pulse rate (SD) | 89.5 (17.5) | 88.7 (17.8) | 90.7 (16.9) | 0.30 | 103.4 (17.7) | 86.7 (16.1) | <0.001 |
| High pulse rate ($\geq$100/min) | 39 (27.1%) | 22 (24.7%) | 17 (30.9%) | 0.42 | 14 (58.3%) | 25 (20.8%) | <0.001 |
| Mean respiratory rate (SD) | 25.3 (7.9) | 23.2 (6.5) | 28.6 (8.7) | <0.001 | 35.5 (9.1) | 23.2 (5.7) | <0.001 |
| High respiratory rate (>22/min) | 63 (44.1%) | 26 (29.2%) | 37 (68.5%) | <0.001 | 22 (91.7%) | 41 (34.5%) | <0.001 |
| Mean oxygen saturation (SD) | 90.7 (11.7) | 91.3 (11.3) | 89.6 (12.5) | 0.60 | 76.3 (18.0) | 93.6 (7.2) | <0.001 |
| Low oxygen level (<94%) | 57 (40.1%) | 33 (37.1%) | 24 (45.3%) | 0.33 | 21 (87.5%) | 36 (30.5%) | <0.001 |
| Low oxygen (<90%) | 34 (23.9%) | 20 (22.5%) | 14 (26.4%) | 0.59 | 18 (75.0%) | 16 (13.6%) | <0.001 |
| Abnormal chest x-ray or CT (n = 15) | 15 (100.0%) | 15 (100.0%) | — | — | 3 (100.0%) | 12 (100.0%) | — |
| Pulmonary infiltrate (n = 15)[2] | 4 (26.7%) | 4 (26.7%) | — | — | 0 (0.0%) | 4 (33.3%) | 0.52 |
| Systemic Inflammatory Response Syndrome (SIRS)[3] | 54 (37.8%) | 28 (31.5%) | 26 (48.1%) | 0.046 | 17 (70.8%) | 37 (31.1%) | <0.001 |
| Acute Respiratory Distress Syndrome (ARDS)[4] | 24 (16.7%) | 14 (15.7%) | 10 (18.2%) | 0.70 | 12 (50.0%) | 12 (10.0%) | <0.001 |
| Sepsis[5] | 6 (4.2%) | 1 (1.1%) | 5 (9.1%) | 0.030 | 4 (16.7%) | 2 (1.7%) | 0.007 |
| **Nutrition (n = 105)[6]** Mean BMI (SD) | 26.5 (6.3) | 26.7 (6.4) | 25.9 (6.3) | 0.78 | 27.1 (6.4) | 26.4 (6.3) | 0.59 |
| Obese | 23 (21.9%) | 18 (22.5%) | 5 (20.0%) | >0.99 | 3 (37.5%) | 20 (20.6%) | 0.23 |
| Overweight | 31 (29.5%) | 23 (28.7%) | 8 (32.0%) | | 3 (37.5%) | 28 (28.9%) | |
| Normal weight | 43 (41.0%) | 33 (41.3%) | 10 (40.0%) | | 1 (12.5%) | 42 (43.3%) | |
| Underweight | 8 (7.6%) | 6 (7.5%) | 2 (8.0%) | | 1 (12.5%) | 7 (7.2%) | |
| Anemia[7] (N = 59) | 16 (27.1%) | 16 (27.1%) | — | — | 2 (25.0%) | 14 (27.5%) | >0.99 |
| **Therapies Provided** IV Fluids | 54 (37.5%) | 37 (41.6%) | 17 (30.9%) | 0.20 | 13 (54.2%) | 41 (34.2%) | 0.065 |
| Oral rehydration | 16 (11.1%) | 9 (10.1%) | 7 (12.7%) | 0.63 | 5 (20.8%) | 11 (9.2%) | 0.150 |
| Non-invasive oxygen support | 57 (39.6%) | 26 (29.2%) | 31 (56.4%) | 0.001 | 22 (91.7%) | 35 (29.2%) | <0.001 |
| **Positioning** Prone | 70 (48.6%) | 60 (67.4%) | 10 (18.2%) | <0.001 | 6 (25.0%) | 64 (53.3%) | <0.001 |
| Cardiac Half-Seated | 52 (36.1%) | 29 (32.6%) | 23 (41.8%) | | 18 (75.0%) | 34 (28.3%) | |
| Other/None | 22 (15.3%) | 0 (0.0%) | 22 (40.0%) | | 0 (0.0%) | 22 (18.3%) | |
| **Medications** Remdesivir | 0 (0.0%) | 0 (0.0%) | 0 (0.0%) | — | 0 (0.0%) | 0 (0.0%) | — |
| Other Antivirals[8] | 11 (7.6%) | 10 (11.2%) | 1 (1.8%) | 0.052 | 2 (8.3%) | 9 (7.5%) | 0.990 |
| Antibiotics | 110 (76%) | 60 (67%) | 50 (91%) | 0.001 | 17 (70.8%) | 93 (77.5%) | 0.480 |
| Antipyretics | 51 (35.4%) | 15 (16.9%) | 36 (65.5%) | <0.001 | 11 (45.8%) | 40 (33.3%) | 0.240 |
| Anticoagulants | 22 (15.3%) | 4 (4.5%) | 18 (32.7%) | <0.001 | 10 (41.7%) | 12 (10.0%) | <0.001 |
| Antimalarials | 18 (12.5%) | 2 (2.2%) | 16 (29.1%) | <0.001 | 2 (8.3%) | 16 (13.3%) | 0.740 |
| Non-steroid Anti-inflammatory | 56 (38.9%) | 6 (6.7%) | 50 (90.9%) | <0.001 | 16 (66.7%) | 40 (33.3%) | 0.002 |
| Corticosteroids | 62 (43.1%) | 14 (15.7%) | 48 (87.3%) | <0.001 | 15 (62.5%) | 47 (39.2%) | 0.035 |
| Vasopressors | 0 (0.0%) | 0 (0.0%) | 0 (0.0%) | — | 0 (0.0%) | 0 (0.0%) | — |
| **Vitamins** Multivitamin | 45 (31.2%) | 0 (0.0%) | 45 (81.8%) | <0.001 | 12 (50.0%) | 33 (27.5%) | 0.030 |
| Vitamin C | 64 (44.4%) | 55 (61.8%) | 9 (16.4%) | <0.001 | 2 (8.3%) | 62 (51.7%) | <0.001 |
| Multivitamin and/or Vitamin C | 102 (70.8%) | 55 (61.8%) | 47 (85.5%) | 0.002 | 13 (54.2%) | 89 (74.2%) | 0.049 |
| Vitamin A | 8 (5.6%) | 0 (0.0%) | 8 (14.5%) | <0.001 | 1 (4.2%) | 7 (5.8%) | >0.99 |

(*Continued*)

**Table 2.** (Continued)

| Clinical Observations | All Inpatients | By Enrollment Country | | | By Outcome | | |
|---|---|---|---|---|---|---|---|
| | | DR Congo | S Sudan | p-value | Deceased | Recovered | p-value |
| | N = 144 | N = 89 | N = 55 | | N = 24 | N = 120 | |
| Zinc | 28 (19.4%) | 17 (19.1%) | 11 (20.0%) | 0.890 | 3 (12.5%) | 25 (20.8%) | 0.410 |

[1]SBP = systolic blood pressure, DBP = diastolic blood pressure

[2]Observed in chest x-ray or CT scan

[3]SIRS defined by two or more of the following: temperature >38.0C or <36.0C, heart rate >90, respiratory rate >20, WBC >12000 or <4000

[4]ARDS defined as acute onset (≤1 week of new/worsening respiratory distress) AND PaO2/FiO2 ≤300 mmHg; OR SpO2/FiO2 ratio ≤315 (adults) or ≤264 (children); AND pulmonary edema not explained by fluid overload or cardiac failure; OR bilateral opacities on chest X-ray/CT scan not explained by effusions, lung collapse or nodule

[5]Sepsis defined as temperature >37.5C or <35.5C AND shock (lethargy, fast breathing, cold skin, prolonged capillary refill, fast weak pulse) AND seriously ill with no apparent cause

[6]Mean Body Mass Index (BMI) or BMI for age; categorized using WHO age/sex specific standards, ages ≥15yrs

[7]Anemia defined by age/sex group as follows: <12 yrs Hb<11g/dL, 12–15 yrs Hb<12g/dL, non-pregnant women ≥15yrs Hb<12/dL and men ≥15yrs Hb<13g/dL.

[8] Including Lopinavir and Ritonavir.

kidney and liver function tests, respectively, were available for only 16.0% (n = 23) and 2.8% (n = 4) of patients. Blood sugar control was assessed for 17 of 28 patients (60.8%) reporting diabetes history; 35.3% (n = 6) had HbA1C levels >8.0, indicating poor diabetes control. There were 61 suspected malaria cases (defined as taking antimalarials and having fever or chills) of which 75.4% (n = 46) had rapid diagnostic tests (RDTs); testing was more common in DRC than SSD (45% vs. 11%) and overall, 15.2% (n = 7) tested positive. When interpreting these findings, it is important to consider the observational nature of the study and the fact that co-morbidities are likely to be under-described, as a result of underdiagnosis, lack of confirmatory findings or inaccurate reporting of medical history.

Deceased patients were significantly more likely to have received non-invasive oxygen support (91.7% vs. 29.2%, p<0.001) within the first 24 hours of admission; one individual in DRC later received bilevel positive airway pressure (BIPAP) or continuous positive airway pressure (CPAP) and subsequently died. Although BIPAP/CPAP machines and ventilators were available in facilities, use was limited by irregular electricity and human resource constraints, including absence of skilled staff and the time-intensive nature of monitoring needs. Deceased patients were more likely to receive non-steroid anti-inflammatories (66.7% vs. 33.3%, p = 0.002), corticosteroids (62.5% vs. 39.2, p = 0.035), and anticoagulants (41.7% vs. 10.0%, p<0.001) at admission, likely a function of a worse clinical condition. Deceased patients were less likely to receive multivitamins or vitamin C (54.2% vs. 74.2%, p = 0.049). The proportions of patients receiving these therapies ever during the hospital stay were slightly greater than at admission, as would be expected. (**S1 Fig**). Of note, only one patient in DRC received vasopressors and three patients in SSD received remdesivir while hospitalized; in DRC, remdesivir was available only in one facility and was a second line treatment.

In regression models estimating adjusted odds ratio (AOR) of survival (**Table 3**), age was the only demographic variable significantly associated with survival, with a 6% survival decrease per additional year (AOR = 0.94, CI:0.91–0.98). With regard to symptoms and health history, survival AORs were significantly reduced among patients with shortness of breath (AOR = 0.16, CI:0.03–0.80), and wheezing (AOR = 0.20, CI:0.05–0.81) and prior diabetes diagnosis (AOR = 0.28, CI:0.09–0.90). Nearly all abnormal clinical observations were significantly associated with decreased survival; these included low oxygen levels (SaO2<94% AOR = 0.10,

## Survival Probability by Country

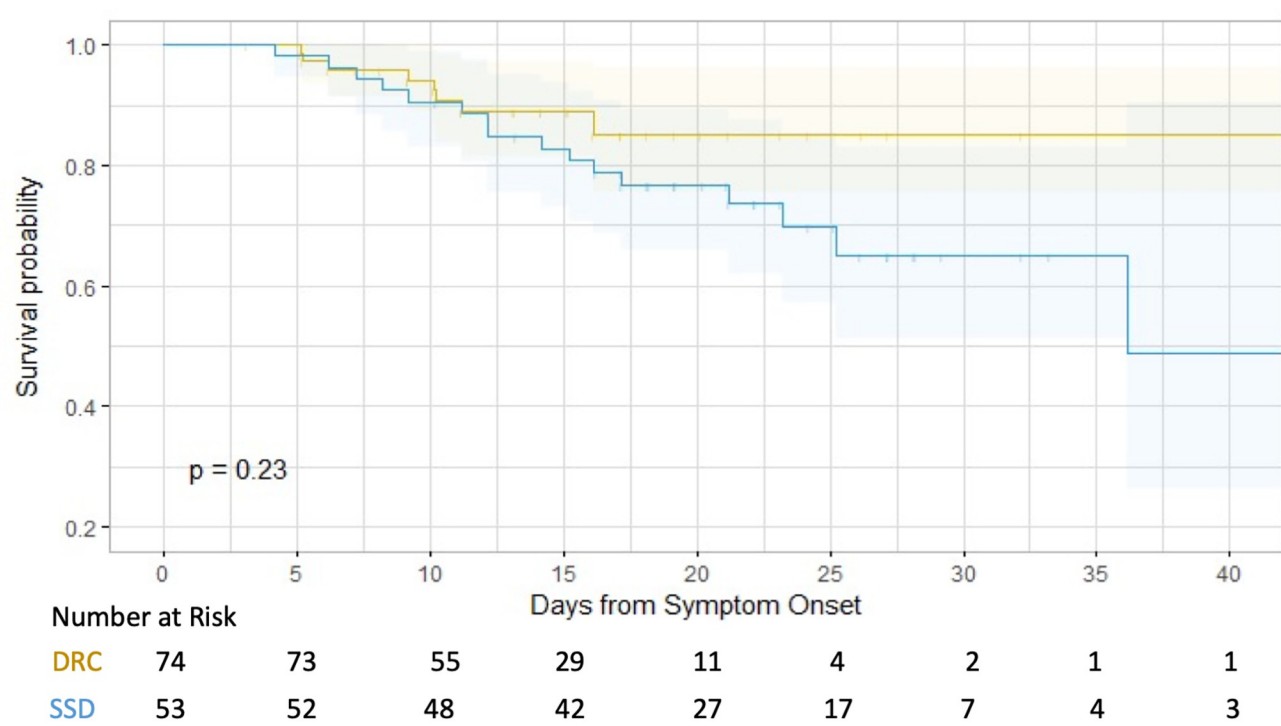

## Survival Probability by Oxygen Level at Admission

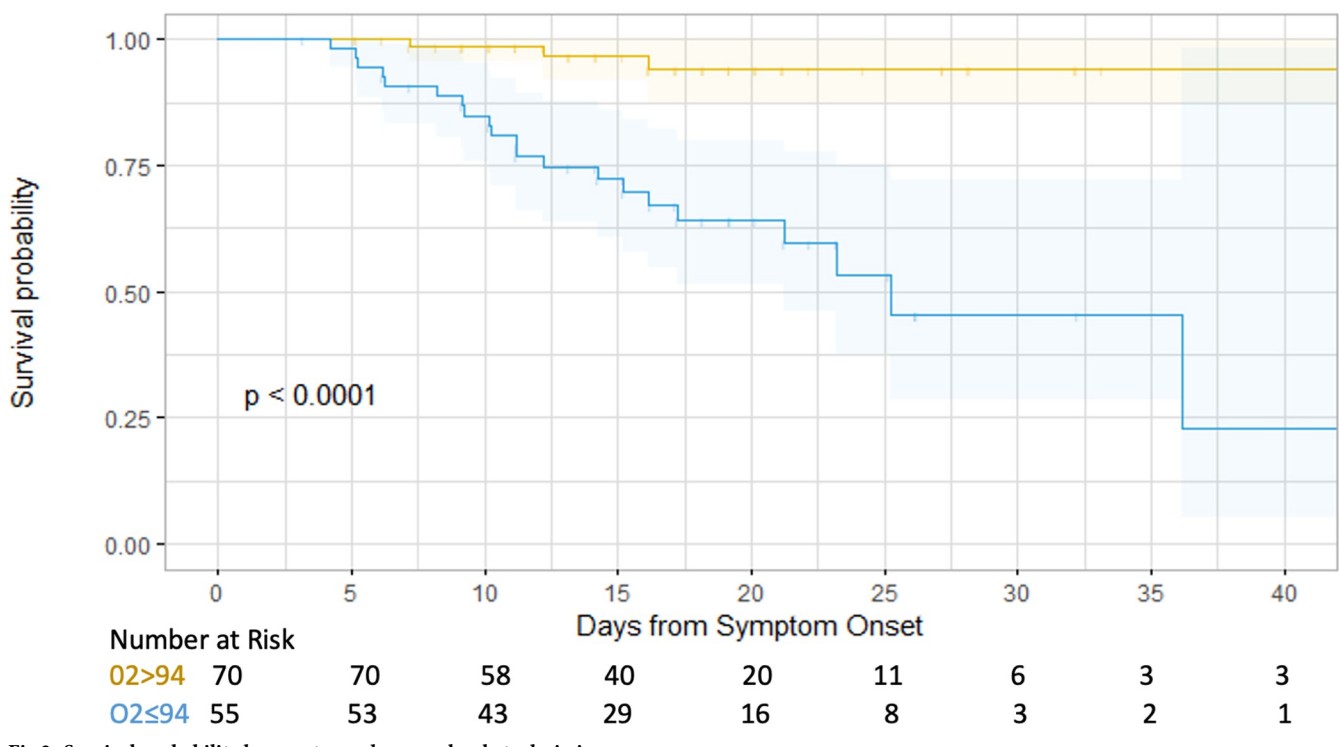

**Fig 2. Survival probability by country and oxygen level at admission.**

**Table 3. Unadjusted and adjusted odds of survival by select patient characteristics and treatments received.**

| | Unadjusted Odds | | | Adjusted Odds[1] | | |
|---|---|---|---|---|---|---|
| | Point Estimate | 95% CI | p-value | Point Estimate | 95% CI | p-value |
| **Demographic Characteristics** | | | | | | |
| Age | 0.94 | (0.92–0.97) | <**0.001** | 0.94 | (0.91–0.98) | **0.004** |
| Male sex (ref: female) | 0.5 | (0.19–1.35) | 0.171 | 0.49 | (0.15–1.57) | 0.229 |
| South Sudan (ref: DRC) | 0.24 | (0.09–0.61) | **0.003** | 0.19 | (0.03–1.43) | 0.107 |
| National (ref: non-nationals) | 1.6 | (0.34–7.51) | 0.55 | 3.08 | (0.57–15.53) | 0.192 |
| **Symptoms and Health History at Admission (self-reported)** | | | | | | |
| Symptom onset to final disposition (days) | 1.10 | (1.02–1.18) | **0.013** | 1.26 | (1.12–1.43) | <**0.001** |
| Fatigue/malaise | 0.26 | (0.08–0.81) | **0.020** | 0.29 | (0.08–1.12) | 0.073 |
| Shortness of breath | 0.07 | (0.02–0.32) | <**0.001** | 0.16 | (0.03–0.80) | **0.026** |
| Chest pain | 0.25 | (0.10–0.65) | **0.004** | 0.55 | (0.18–1.71) | 0.303 |
| Runny nose | 2.48 | (0.86–7.14) | 0.091 | 5.35 | (1.34–21.38) | **0.018** |
| Loss of taste/smell | 0.33 | (0.12–0.9) | **0.030** | 0.79 | (0.24–2.55) | 0.687 |
| Abdominal pain | 5.45 | (0.7–42.48) | 0.105 | 9.05 | (0.97–84.67) | 0.054 |
| Wheezing | 0.27 | (0.08–0.92) | **0.036** | 0.20 | (0.05–0.81) | **0.023** |
| Loss of appetite | 0.17 | (0.04–0.75) | **0.019** | 0.29 | (0.06–1.53) | 0.145 |
| Prior diabetes diagnosis | 0.20 | (0.08–0.51) | <**0.001** | 0.28 | (0.09–0.90) | **0.032** |
| **Clinical Observations at Admission** | | | | | | |
| Fever (>37.5 C) | 0.38 | (0.14–1.00) | **0.051** | 0.27 | (0.08–0.93) | **0.037** |
| Mean pulse rate | 0.95 | (0.92–0.97) | <**0.001** | 0.93 | (0.89–0.96) | <**0.001** |
| High pulse rate (≥100/min) | 0.19 | (0.07–0.47) | <**0.001** | 0.11 | (0.03–0.35) | <**0.001** |
| Mean respiratory rate | 0.83 | (0.78–0.89) | <**0.001** | 0.74 | (0.65–0.85) | <**0.001** |
| High respiratory rate (>22/min) | 0.05 | (0.01–0.21) | <**0.001** | 0.08 | (0.02–0.38) | **0.002** |
| Mean oxygen saturation | 1.14 | (1.07–1.21) | <**0.001** | 1.16 | (1.08–1.23) | <**0.001** |
| Low oxygen level (<94%) | 0.06 | (0.02–0.22) | <**0.001** | 0.10 | (0.02–0.39) | **0.001** |
| Low oxygen level (<90%) | 0.05 | (0.02–0.14) | <**0.001** | 0.04 | (0.01–0.16) | <**0.001** |
| Systemic Inflammatory Response Syndrome (SIRS) | 0.19 | (0.07–0.49) | <**0.001** | 0.14 | (0.04–0.46) | **0.001** |
| Acute Respiratory Distress (ARDS) | 0.11 | (0.04–0.30) | <**0.001** | 0.14 | (0.04–0.46) | **0.001** |
| Sepsis | 0.08 | (0.01–0.49) | **0.006** | 0.18 | (0.03–1.25) | 0.083 |
| **Therapies and Medications Provided (within 24 hours of admission)** | | | | | | |
| Non-invasive oxygen support (ref: none) | 0.04 | 0.01–0.14 | <**0.001** | 0.07 | (0.01–0.38) | **0.002** |
| Prone Position (ref: cardiac half-seated) | 5.65 | (2.15–16.81) | <**0.001** | 2.08 | (0.54–7.96) | 0.285 |
| Anticoagulants | 0.15 | (0.06–0.37) | <**0.001** | 0.56 | (0.18–1.77) | 0.321 |
| Non-steroid Anti-inflammatories | 0.30 | (0.12–0.67) | **0.001** | 0.86 | (0.19–3.83) | 0.839 |
| Corticosteroids | 0.42 | 0.18–0.94) | **0.035** | 2.11 | 0.53–8.36) | 0.287 |
| Vitamin C or Multivitamin | 2.43 | (0.97–6.00) | 0.054 | 3.54 | (0.99–12.67) | 0.052 |
| Vitamin C | 6.00 | (2.01–25.82) | **0.004** | 2.69 | (0.55–13.23) | 0.222 |
| Multivitamins | 0.37 | (0.16–0.86) | **0.019** | 1.44 | (0.34–6.18) | 0.621 |
| **Health Facility Characteristics** | | | | | | |
| Adequate oxygen concentrators/cylinders | 0.21 | (0.01–0.10) | 0.140 | 0.93 | (0.04–22.60) | 0.963 |
| Ventilators available | 0.25 | (0.09–0.63) | **0.004** | 0.20 | (0.02–1.56) | 0.124 |
| Chest x-ray available | 4.00 | (1.6–10.70) | **0.004** | 5.12 | (0.64–40.88) | 0.124 |
| Electrocardiogram available | 4.00 | (1.6–10.70) | **0.004** | 5.12 | (0.64–40.88) | 0.124 |
| Remdesivir available | 0.43 | (0.16–1.09) | 0.084 | 1.08 | (0.04–26.25) | 0.963 |

[1] Individual risk factor models adjusted for age, sex, country of enrollment and (non)national status (fixed effects) and facility (random effect); health facility risk factor models adjusted for age, sex, country of enrollment and (non)national status (fixed effects)

CI:0.02–0.39; SaO2 <90% AOR = 0.04, CI:0.01–0.16); elevated respiratory rate (AOR = 0.08, CI:0.02–0.38), elevated pulse rate (AOR = 0.11, CI:0.03–0.35), fever (AOR = 0.27, CI:0.08–0.93), systemic inflammatory response syndrome (SIRS) (AOR = 0.14, CI:0.04–0.46) and acute respiratory distress syndrome (ARDS) (AOR = 0.14, CI:0.04–0.46).

Non-invasive oxygen was the only treatment significantly associated with survival (AOR = 0.07, CI:0.01–0.38). Prone positioning was significantly associated with increased survival in unadjusted but not adjusted models. Receipt of non-steroidal anti-inflammatories, corticosteroids and anticoagulants, were significantly associated with decreased survival in unadjusted but not in adjusted models. Receipt of Vitamin C or a multivitamin was associated with increased survival with borderline significance in both unadjusted and adjusted models (AOR = 3.52, CI:0.99–12.67). Adjusted odds of survival were also calculated by health facility characteristics and no differences were observed in adjusted models; it is important to note that certain treatments were rationed and odds may not reflect actual access to a treatment/ diagnostic test. Given the small number of health facilities and differences in survival rates between countries, findings should be interpreted with caution, in particular considering that access to these resources was not ubiquitous among individual participants.

## Discussion

This cohort study details clinical progression and outcomes of 144 hospitalized COVID-19 patients in North and South Kivu in eastern DRC and Juba, SSD. The observed hospital mortality proportion of 16.7% is comparable to findings from global systematic reviews of COVID-19 inpatient survival [16, 17] but lower than prior studies in DRC which observed mortality rates of 22% and 32% and were conducted earlier in the pandemic in Kinshasa [18, 19]. However, actual mortality in study facilities was higher due to the inability to enroll patients presenting near death; this figure was not tracked in SSD, however in DRC 24 patients died before being admitted to the study (2.3% of DRC study participants). Higher mortality in SSD (21.9% vs. 9.0%) can be attributed to later presentation and worse conditions at admission. Time from symptom onset to admission has also been shown to be independently associated with survival prior studies in Kinshasa, DRC [18] Late presentation is potentially the result of variety of a factors, including denial, fear and stigma; minimal testing availability; isolation policies; transportation constraints and costs; use of informal providers; and perceptions of poor treatment availability which deter care seeking [20].

Characteristics associated with increased mortality risk included age and prior diabetes diagnoses, which is consistent with known risk factors and similar to ACCCOS study findings which examined outcomes of hospitalized COVID-19 patients in Africa [9, 21, 22] there was no difference in mortality risk by sex. Comparatively few studies have examined infectious comorbidities, and findings from this study did not suggest an association between increased COVID-19 mortality and concurrent malaria or HIV infection or history of tuberculosis. This is potentially a function of inadequate power due to low prevalence and small sample size; other research suggests that HIV/AIDS increases risk for poor outcomes whereas malaria coinfection does not [23–25]. To the best of our knowledge this is the first study to assess underweight as a COVID-19 risk factor and no significant association with mortality was observed, however, only eight subjects had low BMI (seven survived). Anemia was not associated with increased mortality in this study but a meta-analysis has shown anemia to increase risk for poor COVID-19 [26].

Fever was observed in a smaller proportion of patients than anticipated, however, it is important to note that data presented reflects fever observed at enrollment which is distinct from ever having fever during the course of COVID-19. Fever is dynamic and may not have

been elevated at presentation to hospital; for example, other studies of hospitalized COVID-19 patients have reported only 30% of patients with fever [27] and in both SSD and DRC self-medication with antipyretics is common and could have suppressed fever at the time of the observation, thereby resulting in a lower proportion of patients presenting with fever. Patients with low oxygen levels, elevated respiratory rate, SIRS, ARDS and elevated pulse rate and fever at admission had significantly decreased odds of survival which aligns with existing evidence-[28, 29] Non-invasive oxygen was the only therapy or medication significantly associated with decreased survival in both unadjusted and adjusted models; as models were not adjusted for patient severity, lower odds of survival among patients receiving oxygen likely reflects preferential use of oxygen on patients with poorer clinical presentation on admission. Anticoagulants, non-steroid anti-inflammatory drugs and corticosteroids were associated with decreased survival in unadjusted but not adjusted models. Only several study patients received antivirals or vasopressors during their hospital stay, both of which limited ability to do analysis. In DRC, remdesivir was a second line treatment and not all patients were eligible; in both countries, clinicians reported limited supply resulted in rationing medications.

Most evidence on COVID-19 progression and outcomes is from upper-middle and upper-income countries where comorbidity prevalence and clinical management capacities differ greatly as compared to resource poor settings. Expanding the available evidence to understand if and how outcomes vary by context for known risk factors is critical to informing the health response in resource constrained settings. This study suggests that risk factors for inpatient mortality do not differ greatly from those observed in upper-middle and upper-income settings despite the differing population profile of the African humanitarian context and greater infectious disease prevalence. Vitamin C and multivitamins provision to inpatients however pointed to a protective effect (marginally significant), which is consistent with current evidence that indicates provision of Vitamin C is not associated with improved COVID-10 outcomes [30–33] Differences in the health system capacity were striking, where few patients received recommended medications and ventilation, even in facilities where these resources were available. Human resource factors including lack of training to manage patients receiving invasive oxygen and insufficient staff for oversight of resource-intensive therapies were notable limitations in both countries. Other factors that restricted ventilator use were irregular electricity and lack of supplies to support their use.

## Limitations

Weak surveillance systems, low COVID-19 testing capacity and inconsistent information flow from the laboratory coupled with testing hesitancy, a large proportion of travel-related tests and many unreachable cases were factors that influenced the population that was tested and subsequently identified as eligible for the study. Inability to rapidly hire additional study staff during the February/March and a June health worker strike in SSD, along with security issues and the May 22 Mt. Nyiragongo volcanic eruption in DRC also contributed to a smaller sample than planned (n = 1000) and one that is unlikely to be representative of the population in facility catchment areas. The small sample size coupled with low prevalence of selected patient characteristics/treatment use translated to inadequate power to detect significant differences for some variables. Self-reporting of symptoms and comorbidities may have been inaccurate, in particular for conditions with stigma (e.g., HIV/AIDS, TB); relatedly, co-infections may have been underdiagnosed because tests were not ordered, were unavailable (e.g. malaria) or could not be paid for. It was not feasible to record all clinical course details for inpatients, leading to some information being missed; notably quantity of oxygen used was not collected, and temporality of treatments received was not analyzed due to the fact that many deaths occurred

soon after admission. Finally, challenges with timely delivery of equipment intended for the study, particularly in SSD, necessitated the use of alternative equipment and resulted in missing data, most notably on anemia among those enrolled early in the study.

## Conclusions

In this study of hospitalized COVID-19 inpatients in SSD and DRC, the observed mortality proportion (16.7%) was comparable to other findings globally. Age and history of diabetes were the only characteristics measured at presentation associated with decreased survival. Clinical status measures associated with decreased survival included fever, low oxygen level, elevated respiratory and pulse rates, SIRS and ARDS. Patient positioning and receipt of various classes of medications were not associated with differences in survival; the only therapy significantly associated with survival was non-invasive oxygen. Antivirals, vasopressors and invasive oxygen therapies were rarely received despite demonstrated effectiveness in other settings.

To mitigate the challenges of COVID-19 hospital care in resource poor settings, recommendations include strengthening of training of health providers and customizing trainings to reflect availability of specific medications, therapies and operational constraints at the particular facility. Raising provider awareness of factors associated with poor outcomes and providing clear guidance on recommended treatment paths (e.g., flow charts) according to clinical status could reduce the burden on providers and facilitate treatment that is more closely aligned with current best practice. Beyond health facilities, expanded testing capacity and use of appropriate and context specific social behavior change campaigns to reduce the stigma of COVID-19 and improve the perception of available care could facilitate earlier presentation which also would contribute to improved outcomes.

## Supporting information

**S1 Table. Characteristics of health facilities enrolling cases in this study.**
(DOCX)

**S1 Fig. Therapies received during the course of treatment, by outcome.**
(7Z)

## Acknowledgments

We would like to express our appreciation to other members of the study that helped to facilitate this work, including Dugsiye Ahmed, Arsene Baleke, Deepak Kumar, Jennifer Majer and Kelechi Udoh from IMC and Grace Heymsfield and Aimee Summers from CDC. We are immensely grateful for the efforts of the IMC data collection team including Joseph Bongomin and Sinjaro Madol in South Sudan, and Dr. Fiz Mussa Bashizi, Dr. Jean-Marie Kashosi, Numbi Sabiti Cyprien, Matabaro Cirhuza Jean-Paul, Bienvenu Kambale, Dr. Mulinganya Guy and Dr. Matata Ngilima Sebastien in Democratic Republic of Congo (DRC). The study would not have been possible without the support of the Ministry of Health (MoH), in particular the local health authorities in North Kivu and South Kivu provinces in DRC and the Ministry of Health focal Point Dr. Richard Lako in SSD. We also would like to express our gratitude to the NGOs and actors engaged in COVID-19 control in SSD and DRC, in particular those involved in laboratory diagnosis, surveillance and case management staff, for their efforts to ensure timely sharing of case data which facilitated study enrollment. We would like to acknowledge JHSPH faculty member Mija Ververs; Dr. Dayan Woldemichael and Katja Ericson from IMC HQ and IMC operations staff in DRC and SSD. Finally, our deepest appreciation goes to key informants and study participants who volunteered to engage at an especially difficult personal

time; their generosity has furthered our understanding of COVID-19 impacts in some of the most under-served settings globally.

The findings and conclusions in this report are those of the author(s) and do not necessarily represent the official position of the U.S. Centers for Disease Control and Prevention, the U.S. Agency for International Development, the United States Government, or the other authors' respective institutions.

## Author Contributions

**Conceptualization:** Shannon Doocy, Iris Bollemeijer.

**Data curation:** Shannon Doocy, Eva Leidman.

**Formal analysis:** Eva Leidman.

**Funding acquisition:** Shannon Doocy, Iris Bollemeijer.

**Investigation:** Shannon Doocy, Abdou Sebushishe, Eta Ngole Mbong, Kathleen Page.

**Methodology:** Shannon Doocy, Iris Bollemeijer, Eva Leidman, Abdou Sebushishe, Kathleen Page.

**Project administration:** Shannon Doocy, Iris Bollemeijer, Abdou Sebushishe, Eta Ngole Mbong.

**Resources:** Shannon Doocy.

**Supervision:** Iris Bollemeijer, Abdou Sebushishe, Eta Ngole Mbong.

**Validation:** Iris Bollemeijer, Eva Leidman.

**Writing – original draft:** Shannon Doocy, Iris Bollemeijer.

**Writing – review & editing:** Shannon Doocy, Iris Bollemeijer, Eva Leidman, Abdou Sebushishe, Eta Ngole Mbong, Kathleen Page.

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
