## [Decision Letter · Decision Letter 0]

7 Feb 2022

PGPH-D-21-01027

Clinical Progression and Outcomes of Patients Hospitalized with COVID-19 in Humanitarian Settings: A Prospective Cohort Study in South Sudan and Eastern Democratic Republic of the Congo

Dear Dr. Doocy,

Thank you for submitting your manuscript to PLOS Global Public Health. After careful consideration, we feel that it has merit but does not fully meet PLOS Global Public Health’s publication criteria as it currently stands. Therefore, we invite you to submit a revised version of the manuscript that addresses the points raised during the review process.

Thank you for sharing your work with PLOS Global Public Health. Please respond to the reviewer comments paying particular attention to (1) being cautious about drawing conclusions on clinical severity risk factors or the effectiveness of interventions from a small dataset, (2) considering which therapeutic drugs (if any) were widely available at the time in these contexts as well as the appropriateness of Vitamin C provision as a therapy in light of current evidence, (3) placing the admission criteria, study inclusion criteria, operational definition for acutely ill, case definitions, and outcome definitions in the methods section, and (5) discussing the implications of using a small dataset and (4) comparing study findings to those from the few studies conducted in DRC and South Sudan (e.g., Nlandu et al, https://bmcinfectdis.biomedcentral.com/articles/10.1186/s12879-021-06984-x report predictors of mortality among hospitalized patients in Kinshasa). 

In addition, given that this is a clinical investigation, I ask that you consider broadening the authorship specifically to include any Congolese or South Sudanese clinicians who may have given particular insights to the investigators into the management of these patients and the paper itself (perhaps they are already listed as part of the IMC-CDC COVID-19 Research Team?). They may already be represented in the current author list, so please just note this either way when you send your revision. Also, looking at Figure 1, I see a fair proportion of refusals, mostly in South Sudan (~21% of eligible patients for the overall study). It would be useful to discuss how this in the text and how this impacted the current study of hospitalized patients.

We look forward to receiving your revised manuscript.

Kind regards,

Ruwan Ratnayake, MHS

Academic Editor

Journal Requirements:

2. Please provide  separate figure files in .tif or .eps format only and remove any figures embedded in your manuscript file.  Please ensure that all files are under our size limit of 20MB.  

For more information about how to convert your figure files please see our guidelines: Once you've converted your files to .tif or .eps, please also make sure that your figures meet our format requirements

3. In the online submission form, you indicated that your data will be submitted to a repository upon acceptance.  We strongly recommend all authors deposit their data before acceptance, as the process can be lengthy and hold up publication timelines. Please note that, though access restrictions are acceptable now, your entire data will need to be made freely accessible if your manuscript is accepted for publication. This policy applies to all data except where public deposition would breach compliance with the protocol approved by your research ethics board. If you are unable to adhere to our open data policy, please kindly revise your statement to explain your reasoning and we will seek the editor's input on an exemption. Please be assured that, once you have provided your new statement, the assessment of your exemption will not hold up the peer review process.

4. Please amend your detailed Financial Disclosure statement. This is published with the article, therefore should be completed in full sentences and contain the exact wording you wish to be published.

ii). State the initials, alongside each funding source, of each author to receive each grant.

iii). State what role the funders took in the study. If the funders had no role in your study, please state: “The funders had no role in study design, data collection and analysis, decision to publish, or preparation of the manuscript.”

Reviewers' comments:

Reviewer's Responses to Questions

**Comments to the Author**

1. Does this manuscript meet PLOS Global Public Health’s publication criteria? Is the manuscript technically sound, and do the data support the conclusions? The manuscript must describe methodologically and ethically rigorous research with conclusions that are appropriately drawn based on the data presented.

Reviewer #1: Partly

Reviewer #2: Yes

Reviewer #3: Yes

2. Has the statistical analysis been performed appropriately and rigorously?

Reviewer #1: Yes

Reviewer #2: Yes

Reviewer #3: Yes

3. Have the authors made all data underlying the findings in their manuscript fully available (please refer to the Data Availability Statement at the start of the manuscript PDF file)?

Reviewer #1: No

Reviewer #2: Yes

Reviewer #3: Yes

4. Is the manuscript presented in an intelligible fashion and written in standard English?

Reviewer #1: Yes

Reviewer #2: Yes

Reviewer #3: Yes

5. Review Comments to the Author

Reviewer #1: The authors have reported an interesting study on outcome of COVID-19 patients hospitalized in context of humanitarian setting. However, this study need to be improving in regard of these following remarks.

Introduction

1. Why the authors highlighted the remdesevir treatment in the introduction section given that there is no specific treatment against COVID-19?

Methods

1. The authors did not clearly define the inclusion and exclusion criteria for the study

2. The authors did not clearly define the COVID-19 case and how the diagnostic was made? On base of serological test? Antigenic test? Or RT-PCR?

3. The authors must specify how the sampling of the study was done? Has a sample size been calculated?

4. Operational definition must be clearly presented in the method section

Results

1. Unlike many studies, fever was not a major symptom in this study. How the authors can explain it?

2. How the authors can explain a high proportion of lack of laboratory test in the prospective study context?

3. The authors reported in results section that only one patient in DRC received vasopressor treatment but this information doesn’t appear in table 3.

4. Can the authors provide a staging of patient by OMS COVID-19 classification?

5. Can the authors provide an operational definition for acutely ill?

6. How the authors can explain the fact that some acutely ill patients were treated ambulatory?

7. In the table 3, the authors must specify for hypotension and hypertension the variable systolic blood pressure (SBP) and diastolic blood pressure (DBP)

8. Which other antiviral was used in DRC COVID-19 patients? (table 3)

Discussion

1. The mortality rate reported in this study is very low compared to other study specially in DRC. The authors must discuss this discrepancy.

2. The authors have justified the high mortality in SSD by the later presentation and worse conditions at admission of SSD patients. Can the authors specify if the criteria of admission in the medical center was the same in DRC and Sudan?

3. Only 18.8% of patients have presented fever. Can the authors discuss this results?

4. The authors can add a DRC studies on COVID-19 especially in the ACCCO study, no study from DRC or Sudan was included. Here two studies from DRC: Nlandu, Y., Mafuta, D., Sakaji, J. et al. Predictors of mortality in COVID-19 patients at Kinshasa Medical Center and a survival analysis: a retrospective cohort study. BMC Infect Dis 21, 1272 (2021). https://doi.org/10.1186/s12879-021-06984-x; Makulo, JR., Mandina, M.N., Mbala, P.K. et al. SARS-CoV2 infection in symptomatic patients: interest of serological tests and predictors of mortality: experience of DR Congo. BMC Infect Dis 22, 21 (2022). https://doi.org/10.1186/s12879-021-07003-9

Reviewer #2: Thank you for this work. I think it is really important and I am glad to see this experience is being documented. I am sorry if the comments are many but if there are multiple papers coming from this work, I thought it might be worth discussing some of the issues in more detail, if possible. I see the strength of this paper as documenting COVID in populations where little has been documented. We need greater insight into clinical presentations and severity in different settings. I would be cautious on drawing conclusions on clinical severity risk factors or the effectiveness of interventions from a small dataset, as the epidemic has been impacted so much correlations that did not prove as meaningful with further analysis.

I think it’s really important the paper discusses underweight status. Being overweight has been such a clear risk factor, but in other settings low weight is so often correlated with other factors (say cancer) that it is hard to tease out. I am really glad to see this discussed as a factor as 2 years into the epidemic it has not had the attention and analysis it should.

Abstract

Can the Findings abstract be more specific?

"The observed mortality proportion among hospitalized patients was 16.7% (CI: 11.2-23.3%) and was greater

in South Sudan, where patients presented later after symptom onset and in worse conditions."

Maybe list specific mortality or say, rate is over 3x higher in South Sudan

Interpretation

Page 4

Endnote error in first line.

Was pregnancy assessed? This is certainly a risk factor that causes a substantial loss of life years. Non-pregnant women often have a better survival than men but pregnant women have worse outcomes and in this context would presume would be a frequent enough risk factor.

https://jamanetwork.com/journals/jamapediatrics/fullarticle/2779182

Seroprevalences. it would be good to cite existing data on these two contexts. serotracker does not include good studies on either area but there are more recent studies

Juba

https://pubmed.ncbi.nlm.nih.gov/34013872/

https://www.ajtmh.org/view/journals/tpmd/104/4/article-p1526.xml

https://www.panafrican-med-journal.com/content/article/38/93/full/

This would be important to document for future readers in a world where past infection is common and immunity based on past infections and vaccines will shape a different global epidemic.

Remdesivir (page 4) is commonly used in the US but is not recommended in many countries because of perceived lack of utility. The WHO maintains a conditional recommendation against use of remdesivir. https://www.who.int/publications/i/item/WHO-2019-nCoV-therapeutics-2022.1

I might just stress that evidence based, targeted therapeutics were not available. When this paper will be read, monoclonal antibodies and Paxlovid will be the drugs for which access inequities will be the concerns. There are many in the WHO and other organizations and countries that do not feel the impact of remdesivir is significant enough (there’s a lot of different studies with different marginal findings).

Page 5

Duration of symptoms before presentation: were there any expected delays related to the SSD center being a COVID referral center? Were there extra hurdles to presentation because of this? (This is a common issue in outbreaks where infectious diseases are treated outside of the main hospital system). Were there any other reasons documented why there were delays? (I see later the discussion on denial and other issues, anything more that can document this would help). It would be interesting to have insight into this as this may play a substantial role in affecting mortality.

Page 6

Is it possible to quantify oxygen use? Concentrators vs tanks? Amount of total oxygen that can be delivered? COVID requires high quantities of oxygen and in low resource settings this is often the real limitation. It’s hypoxia (oxygen) and not ventilation (help breathe) that is the main barrier. As such, CPAP is not used that much for COVID even when available and IPC concerns don’t limit use. Rather high flow (like say 60L/min) of oxygen is preferred, even when ventilators and cpap is available. This can be created with oxygen tanks in low resource settings (stringing together multiple tanks). Concentrator machines cannot provide the same amount of oxygen that tanks can; maybe at most 5L/min and of lower percentages of oxygen (80-90%, rather than 98% or more from a tank that reads 5 L/min)

How many tanks of oxygen available per person? What oxygen use rates?

In low resource settings where this very expensive resource is so often limited, there is often a tendency to conserve to avoid running out so patients aren’t able to reach the levels we would have rathered.

Is there monitoring of oxygen and standardized protocols for use? Are oxygen levels kept above certain levels? Are there thresholds at which futility is reached?

Oxygen in low resourced settings is quite expensive, in part because of resource limitations but also monopolies, https://www.yahoo.com/now/lack-oxygen-leaves-patients-africa-080052014.html

and especially in South Sudan this is a major limitation to care. First plant came online in Sept. https://www.afdb.org/en/news-and-events/press-releases/south-sudan-countrys-first-oxygen-plant-comes-stream-juba-hospital-help-fight-covid-19-45393

Concentrator machines were often used for COVID in South Sudan and this would be a major limitation on the extent of care that can be delivered.

Do we know anything about deaths that occurred without oxygen being given? Those who die largely require oxygen so would clarify if had any other info on those deaths with no oxygen. Was it available? Were there lapses in monitoring? There can be some cases of sudden clots but in general almost everyone, where there is close monitoring and available resources, would need oxygen.

Is it known if steroids were given before patients needed oxygen? Steroids have been found to benefit those who have severe COVID but are considered a risk for those without severe COVID or not needing oxygen. The WHO has a “conditional recommendation against systemic corticosteroids in patients with non-severe COVID-19” The risk is of worsened diabetes, secondary infections, and all the side effects of steroids without the benefit. In many low resource settings, steroids have been given out to those who do not need oxygen (O2 is not <90 or not <94, there are different cut offs) in order to do something, but may be causing more harm than good.

https://www.who.int/publications/i/item/WHO-2019-nCoV-therapeutics-2022.1
https://www.thelancet.com/article/S2213-2600(20)30530-0/fulltext

Also, vasopressors given without close monitoring are more dangerous than helpful. They aren’t normally lifesaving interventions without greater clinical capacity.

Because hypoxia alone is the major driver of severe disease along with clots, vasopressors are not as crucial here as in say cases of bacterial sepsis. They may help with bacterial superinfections or in patients who are intubated and sedated, but they are only adjunctive treatments to bridge a patient.

Page 7

Is there any quantification of how many patients arrived near death? Total death figures? Arrival at or near death is of course a common and under-reported factor affecting the reporting of mortality of many outbreaks and so anything to quantify it or total death rates could help.

Is there another paper besides #17 that can be cited as the manuscript is not available? It’s an important topic.

It is really important that paper shows a finding on underweight. Since the beginning of the epidemic, this has been a question with few answers or focus. This is great to see, even though the number (8) enrolled was small. I might stress the need to study this further.

I would edit and qualify further the statement about co-infections. It’s not just that the numbers are low, the diagnostic accuracy is also low. Given the low rates of testing for malaria, it is hard to make a conclusion about malaria, especially as treatment for malaria did not correspond to RDT+ tests. Moreover, as malaria infection can correlate with age and other risk factors like pregnancy, it would need a larger dataset to make any clear conclusions, even if we knew precisely who had malaria and who did not. (If interested, this article collected a number of different studies on malaria and COVID though I think they jumped to some conclusions without clear basis: https://bmcinfectdis.biomedcentral.com/articles/10.1186/s12879-022-07064-4 ) The rates of reported past TB are quite low relative to what would be expected (ie if 1 in 300 develop the disease in a year a past history of TB of 1.1% is quite low). So I am not sure we can feel confident that we have a good grasp on the effects of these diseases. There is a very robust literature on HIV and COVID. There are many who do quite well with COVID who have HIV, even though overall HIV increases risk, so I would not expect to necessarily see an effect in this sample size.

I would cite further sources if discussing risk of anemia in COVID. Some have found it to increase mortality. Others have seen the opposite, higher red blood cell counts, to be associated with mortality (disease does cause clotting). It likely also is often a correlative factor and say a marker for other risks (men/testosterone, pregnancy, sickle cell, cancer etc) so can easily vary in different populations.

https://bmjopen.bmj.com/content/11/2/e044618.long

Please edit the statement "Vitamin C and multivitamins provision to inpatients however

pointed to a protective effect (marginally significant), which is consistent with several studies,28,29 though evidence to date is inconclusive.30,31" Vitamin C was not significant here (nor was it with MVI) in the adjusted Odds table and I would be very careful with trends or values bordering significance. Any statements on therapies that have been studied and routinely shown not to work should be carefully worded. There are many reasons why people receiving one therapy may improve without a directly causal relationship; there may be other factors (like attention to detail or ability to swallow extra meds, which may not be captured in this study). Vitamin C and multivitamins have after many studies not been shown to be effective at this time. There is certainly a difference in vitamin C in different populations, like as Mija Ververs has shown in young male refugees from South Sudan and certainly it would be interesting to see if provision does make a difference, but this would be a different study. However, study after study has shown that they have not been effective. These studies have been cherrypicked by those who do not believe in vaccines to offer alternatives like cocktails of ivermectin, HCQ, and vitamin C, allowing patients to blame doctors and nurses and not those advocated against vaccination. These faux treatments have also been sent to those with fewer resources who do not have access to medical care. Hence I’m always nervous when seeing Vit C described as marginally significant. In fact, papers cited as 28 and 29 actually are studies which have shown that vitamin C has not been effective. Source 29 states as conclusion: "We did not find significantly better outcomes in the group who were treated with HDIVC in addition to the main treatment regimen at discharge." Source 28 states "This pilot trial showed that HDIVC failed to improve IMVFD28, but might show a potential signal of benefit in oxygenation for critically ill patients with COVID-19 improving PaO2/FiO2 even though.”

Human resources is discussed. Was there any impact of when someone presented? Any impact of the strike on mortality? Were there clinical surges where mortality or other outcomes was affected? Was there any difference in mortality over time?

Looking back on this epidemic, it will be important to clarify vaccination rates (as it may not be clear to future readers how non-existent vaccination was well into 2021) and the impact of variants and past infection (seroprevalence) on outcomes. Mortality will be different in different phases of the epidemic as we continue on, given differences in variants and immunity. So it would help readers in the future if documenting any info on surges and variants and specifics on vaccine inequity.

It appears generally that in March 2021 there was a surge in South Sudan, while Jan and July 2021 surges in DRC and Rwanda. There is some documentation on variants found, but not a lot of course in these contexts, so certainly do not need to find what cannot be found, but may make mention of changing variants worldwide.

https://radiotamazuj.org/en/news/article/covid-19-south-sudan-witnesses-resurgence-confirms-delta-variant

Table 1

Can you provide info on the size of the catchment area? Distance for travel?

North and South Kivu are obviously much larger than Juba. What populations or geographic areas do these clinics cover?

Would change “Clinical staff trained” to “Staff had COVID training” or something similar. Thee footnote may not be read and I don’t want readers thinking there are untrained doctors and nurses.

Are all the nurses and doctors specifically RNs and MDs or are there also nursing assistants and CO’s or other providers? Was there a dedicated number of providers from the mixed use facilities who worked with COVID patients.

Might not say N95 equivalents, but just say surgical masks, KN94, or KN95 or what was available. Is there a sense of how many had surgical masks and how many N95s? Just better for future documentation to understand.

Did patients have continuous pulse oximetry? Any info on how regularly pulse ox was checked?

Is there any info on antibiotics used and how frequently?

Table 2

Those who did not have symptoms but were admitted, any idea what admission for? Were these admitted for non-COVID diagnoses?

Would stress that Self-reported prior tuberculosis (despite heading) as these numbers are lower than expected. Some studies use CXR criteria for prior TB so it’s good to add self-reported.

Definition of ARDS has an “OR” where there should be an “AND” It should also be 3 and not 2

ARDS defined as acute onset (≤1 week of new/worsening respiratory distress) AND PaO2/FiO2 ≤300

mmHg OR SpO2/FiO2 ratio ≤315 (adults) or ≤264 (children); AND pulmonary edema not explained by fluid overload or cardiac failure; OR

bilateral opacities on chest X-ray/CT scan not explained by effusions, lung collapse or nodule;

Make ARDS 3 and then change the other numbers so the footnotes line up. There are two footnotes labelled as 2 (SIRS and ARDS) and this means the footnotes don’t line up.

Figure 3 shows a very high use of antibiotics. Only 3-7% of COVID cases on admit are found to have coinfections with bacteria. There is certainly a lot of overuse of antibiotics which, if not benefiting pt, is exposing them to harm (diarrhea, AMR, renal/hepatic/allergic/cardiac side effects). If possible, I would explore antibiotic use more. Is there any access in the higher resource clinics to micro testing? Microbiology lab? (clearly not in the lower resourced SSD clinic but perhaps in the DRC clinic with a CT scanner). Bacterial superinfections occur much more commonly with intubation which unfortunately was not a possibility here.

Reviewer #3: The manuscript is technically sound and uses sound data to support the conclusions. However, there are some typographical errors:

- Under the introduction: the last line on paragraph 2; the use of the word 'concerns' should be revised. I would suggest it is replaced with 'inadequate'.

- Under the methods: the second last line on paragraph 1, I suggest that the sentence extract, '.....a secondary aim was to document clinical progression and characterise COVID-19 management' is rewritten. This sentence extract should be rewritten as '...a secondary aim was to document clinical progression and characterise COVID-19 clinical management'.

Under the results: I suggest that the sentence, 'While the mortality proportion differed significantly between SSD and DRC overall, there was no difference in survival probability when adjusted for from the date of symptom onset', be rewritten. The sentence should be rewritten as, 'While the mortality proportion differed significantly between SSD and DRC overall, there was no difference in survival probability when adjusted for the date of symptom onset'.

**
General remarks 
**

The paper presented clinical profile of and outcomes of admitted COVID-19 patients in five health facilities in the Democratic Republic of Congo and South Sudan – both in sub-Saharan Africa. As reported elsewhere, the study showed COVID-19 survival is influenced by presence of underlying illnesses like diabetes as well as timely access to quality care right from the time the symptoms of illness commence. These findings are therefore critical and justify the need to prop up COVID-19 surveillance and case management to avert adverse outcomes and improve survival. The manuscript is technically sound and uses sound data to support the conclusions.

**
Specific comments and suggestions  
**

Please state how the sample size was determined for the study. Alternatively, if these were the total number of patients that could be enrolled, it is critical that a power calculation is done ensure that the study was adequately powered. If the power determination reveals the study was not adequately powered, then the limitations should be updated accordingly.It is important that the authors state the case definition for COVID-19 cases that were included in the study. This is critical for assessing the role of misclassification bias in the study.Similarly, the authors should provide the definitions for the outcomes (COVID-19 recovery and COVID-19 death) used in the study. Clear definitions and objective determination of the outcomes are critical for minimizing the risk of bias in the assessment of the outcome.Similarly, the admission criteria for patients should be stated. It is also possible that these criteria differed in the two countries where the study was undertaken given the country-specific capacities for managing COVID-19 patients. How were these county and pandemic phase specific changes and differences in admission criteria for facility based care adjusted for in the study?Besides malaria and self-reported diabetes, please state how the presence of other underlying diseases like HIV, tuberculosis, and other non-communicable diseases were assessed among cases enrolled into the study?

6. PLOS authors have the option to publish the peer review history of their article (what does this mean?). If published, this will include your full peer review and any attached files.

**Do you want your identity to be public for this peer review?** For information about this choice, including consent withdrawal, please see our Privacy Policy.

Reviewer #1: No

Reviewer #2: No

Reviewer #3: **Yes: **Joseph Francis Wamala

---

## [Decision Letter · Decision Letter 1]

27 Jul 2022

Clinical Progression and Outcomes of Patients Hospitalized with COVID-19 in Humanitarian Settings: A Prospective Cohort Study in South Sudan and Eastern Democratic Republic of the Congo

PGPH-D-21-01027R1

Dear Dr Doocy,

We are pleased to inform you that your manuscript 'Clinical Progression and Outcomes of Patients Hospitalized with COVID-19 in Humanitarian Settings: A Prospective Cohort Study in South Sudan and Eastern Democratic Republic of the Congo' has been provisionally accepted for publication in PLOS Global Public Health.

Best regards,

Ruwan Ratnayake

Academic Editor

Thank you for your work on improving this manuscript. We are happy to accept it.

Reviewer 2 has given some helpful suggestions (and observations). I would emphasize point 1 (on stating the timeline) and point 5 (on commenting on the diabetes finding) as being the most important to think about in terms of updating the manuscript. Again, this is up to you, and you can return the manuscript without these points incorporated as well. Reviewer 1 requested that you fix a reference. Please see below for all the comments.

**Comments to the Author**

1. If the authors have adequately addressed your comments raised in a previous round of review and you feel that this manuscript is now acceptable for publication, you may indicate that here to bypass the “Comments to the Author” section, enter your conflict of interest statement in the “Confidential to Editor” section, and submit your "Accept" recommendation.

Reviewer #1: All comments have been addressed

Reviewer #2: All comments have been addressed

Reviewer #3: All comments have been addressed

2. Does this manuscript meet PLOS Global Public Health’s publication criteria? Is the manuscript technically sound, and do the data support the conclusions? The manuscript must describe methodologically and ethically rigorous research with conclusions that are appropriately drawn based on the data presented.

Reviewer #1: Yes

Reviewer #2: Yes

Reviewer #3: Yes

3. Has the statistical analysis been performed appropriately and rigorously?

Reviewer #1: Yes

Reviewer #2: Yes

Reviewer #3: Yes

4. Have the authors made all data underlying the findings in their manuscript fully available (please refer to the Data Availability Statement at the start of the manuscript PDF file)?

Reviewer #1: Yes

Reviewer #2: Yes

Reviewer #3: Yes

5. Is the manuscript presented in an intelligible fashion and written in standard English?

Reviewer #1: Yes

Reviewer #2: Yes

Reviewer #3: Yes

6. Review Comments to the Author

Reviewer #1: The authors have adequately addressed The comments raised in our previous round of review and the manuscript is now acceptable for publication.

The reference 17 should still be corrected: Makulo JR, Mandina MN, Mbala PK, Wumba RD, Akilimali PZ, Nlandu YM, Odio JO, Bepouka BI, Longokolo MM, Mukenge EK, Kamwiziku G, Muamba JM, Longo AL, Lufu CM, Keke HL, Mbula MM, Situakibanza HN, Sumaili EK, Kayembe JN. SARS-CoV2 infection in symptomatic patients: interest of serological tests and predictors of mortality: experience of DR Congo. BMC Infect Dis. 2022 Jan 4;22(1):21. doi: 10.1186/s12879-021-07003-9. PMID: 34983411; PMCID: PMC8724652.

Reviewer #2: My points are minor and only to ensure clarity.

1.

In the intro, I would be clear on the time frame, so it is easy for others read in the future when the phases of the early COVID pandemic are a blur. I would maybe add "as of 2021" to this sentence in the intro:

Most African countries have low vaccination coverage, inadequate diagnostic and laboratory capacity,

and at hospitals limited staffing and availability of evidence-based COVID-19 treatments such as

ventilators, antivirals and monoclonal antibodies hamper quality of care.

2.

I would clarify about hospitalized on enrollment versus hospitalized after enrollment. This takes a moment to decipher so I would just add a few words. From what the paper states, the 751 were recruited as outpatient or inpatient and 144 were enrolled as hospitalized patients and were followed to death or recovery. Is there any data on those enrolled as outpatients who were later admitted? The description used in the abstract stated they were enrolled as hospitalized patients, so were there others who were enrolled as outpatients and later hospitalized? how many?

ie

All eligible cases (n=751) were recruited of which 592 consented to

participate and were enrolled and 519 were followed to recovery or death, including 144 patients

hospitalized at four health facilities which are the focus of this paper (Figure 1)

3. Is there any data on transfers? reasons? status? were these patients sicker? had more resources? were pregnant or in labor and sent to another facility? etc Understandable if not known. Just sometimes transfer is clearly impacting outcomes. No reason to search further for more info on this. Since 24 deceased and 5 transferred, if all critically ill may be more pertinent, so just curious if any more info on these.

4. Curious for admission criteria among those whose O2 sat was >94% on admit. O2 <94 is a common cut off for admission in many settings. It may show more caution in a lower resourced setting or may represent other factors in clinical decision making. With PaO2 levels may also have some discrepancy on SpO2 reads depending on monitor and also skin tone. Again, no reason to change anything, just curious.

5. Nothing for you to add for #5. I just think in the discussion that it's an interesting finding that diabetes plays such a role here. It certainly is an independent risk factor and often tracks with other known risk factors - age, weight, cardiovascular disease, renal disease. It is also interesting in respect steroid use, which may be more harmful in low resource settings where frequent glucose monitoring is not feasible (ie diabetes and steroids can lead to dangerously high blood sugars but also create more risk for mucor and other fungal infections which may not be recognized and may differ by geography). This concern has been raised in settings like India as potentially a factor worsening outcomes as the treatment (steroids) was studied and found to be beneficial in settings with closer clinical monitoring.

It may also be that those with undiagnosed diabetes remain at even greater risk.

Likewise anticoagulants which were popular earlier in the pandemic have been found to not improve outcomes as much as hoped when no clots identified (and identifying such clots in low resource settings is of course much harder).

This all just speaks to why we need trials to monitor the safety of interventions of novel and emerging diseases in lower and higher resourced settings. These interventions may require clinical monitoring and course adjustment, which are considered part of basic care in higher resourced settings, but which may be out of reach in settings with resources are strained- both physical and Human Resources.

That said, we simply don't have the trials or evidence to know. We just need more data and we do know more advanced treatment is often a marker of more severe illness. So hopefully this work here will inspire more data collection in future outbreaks (and in COVID as well) in lower resourced settings.

Thank you for all of this work. It will definitely help all of us.

Reviewer #3: no additional comments.

7. PLOS authors have the option to publish the peer review history of their article (what does this mean?). If published, this will include your full peer review and any attached files.

**Do you want your identity to be public for this peer review?** For information about this choice, including consent withdrawal, please see our Privacy Policy.

Reviewer #1: No

Reviewer #2: No

Reviewer #3: **Yes: **Joseph Francis Wamala
